# Evolutionary association of receptor-wide amino acids with G protein–coupling selectivity in aminergic GPCRs

Berkay Selçuk[1], Ismail Erol[2,3], Serdar Durdaği[2], Ogün Adebali[1,4]

**G protein-coupled receptors (GPCRs) induce signal transduction pathways through coupling to four main subtypes of G proteins (G_s, G_i, G_q, and G_{12/13}), selectively. However, G protein selective activation mechanisms and residual determinants in GPCRs have remained obscure. Herein, we performed extensive phylogenetic analysis and identified specifically conserved residues for the aminergic receptors having similar coupling profiles. By integrating our methodology of differential evolutionary conservation of G protein–specific amino acids with structural analyses, we identified specific activation networks for G_s, G_{i1}, G_o, and G_q. To validate that these networks could determine coupling selectivity we further analyzed G_s-specific activation network and its association with G_s selectivity. Through molecular dynamics simulations, we showed that previously uncharacterized Glycine at position 7x41 plays an important role in receptor activation and it may determine G_s coupling selectivity by facilitating a larger TM6 movement. Finally, we gathered our results into a comprehensive model of G protein selectivity called "sequential switches of activation" describing three main molecular switches controlling GPCR activation: ligand binding, G protein selective activation mechanisms, and G protein contact.**

## Introduction

G protein–coupled receptors (GPCRs) constitute a significant group of membrane-bound receptors that contain five different classes (Fredriksson et al, 2003; Rosenbaum et al, 2009). The aminergic subfamily of receptors is present in class A and includes receptors for dopamine, serotonin, epinephrine, histamine, trace amine, and acetylcholine (Vass et al, 2019). With a large number of known coupling profiles, experimental structures, and mutagenesis experiments available, aminergic receptors are by far the most studied subfamily of GPCRs. These receptors can couple with different heterotrimeric G proteins which induce distinct downstream signaling pathways (Wettschureck & Offermanns, 2005). Disruption

of the proper receptor activation is likely to be the cause of diseases such as coronary heart disease (Wang et al, 2018a) or major depression (Catapano & Manji, 2007; Senese et al, 2018). Therefore, understanding the molecular mechanisms of coupling selectivity is crucial for developing better therapeutics and diagnostics.

With the advancement of new methodologies, two recent studies have revealed the G protein–coupling profiles of a large set of receptors. Inoue et al (2019) have used a shedding assay-based method to measure chimeric G protein activity for 11 unique chimeric G proteins representing all human subtypes and 148 human GPCRs. Because they have not managed to find an evident conserved motif determining G protein selectivity between receptors, they have built a machine learning-based prediction tool to identify sequence-based important features for each G protein. Similarly, Avet et al (2022) have used a BRET-based method detecting the recruitment of the G protein subunits to the receptor to reveal coupling profiles for 100 different receptors. The main strength of this study is that it does not require a modified G protein. Although both high-throughput studies largely agree with each other for certain G proteins, there are inconsistencies between the datasets. Thus, these valuable resources should be analyzed together in detail to gain more power in identifying the selectivity-determining factors in G protein coupling.

Several attempts have been made to identify molecular determinants of G protein coupling. Most of these (Chung et al, 2011; Semack et al, 2016; Du et al, 2019; Liu et al, 2019; Okashah et al, 2019) have focused on the G protein–coupling interface by analyzing contacts between receptor and the G protein. The others (Rose et al, 2014; Kang et al, 2018; Van Eps et al, 2018; Wang & Miao, 2019) have highlighted the structural differences between receptors that couple to different G proteins. Flock et al (2017) have analyzed the evolutionarily conserved positions of orthologous and paralogous G proteins and proposed the "lock and key" model. According to their model, G proteins (locks) have evolved with subtype-specific conserved barcodes that have been recognized by different subfamilies of receptors (keys). Because receptors with distinct evolutionary backgrounds can couple to the same G protein, receptors also must have evolved to recognize the existing barcodes. Although the model has explained the selectivity-determining

[1]Molecular Biology, Genetics and Bioengineering Program, Faculty of Engineering and Natural Sciences, Sabanci University, Istanbul, Turkey  [2]Computational Biology and Molecular Simulations Laboratory, Department of Biophysics, School of Medicine, Bahcesehir University, Istanbul, Turkey  [3]Department of Chemistry, Gebze Technical University, Gebze, Turkey  [4]TÜBiTAK Research Institute for Fundamental Sciences, Gebze, Turkey

Correspondence: oadebali@sabanciuniv.edu

interactions between G protein and receptors, we still lack subfamily-specific receptor signaling mechanisms that involves but not limited to the G protein-coupling interface.

Despite the extensive research carried out to identify the determinants of G protein selectivity, selectivity-determining positions within receptors has remained underexplored. Here, we developed a novel methodology to identify a set of specifically conserved residues for the receptors sharing similar coupling profiles through the identification of orthologous receptors. Structural analyses revealed that specifically conserved positions are part of G protein–specific activation pathways that allow receptors to transduce the signal from the ligand-binding pocket to the G protein–coupling interface, induce the necessary conformational changes to get coupled by the relevant G protein subtype.

# Results

After a gene duplication event, paralogous clades might diverge from each other with respect to their functions. Therefore, evolutionary pressure against paralogous genes might differ. To perform a precise conservation analysis, we aimed to identify the gene duplication nodes in aminergic receptor evolution. We identified receptor subfamilies (orthologous and paralogous sequences) through a meticulous phylogenetic analysis. As we previously proposed (Adebali et al, 2016), the variations that observed in a paralog protein of interest may not be tolerated in the orthologous proteins. In our analyses, we only used orthologous receptors to define a subfamily of interest, members of which are likely to retain the same function. This approach greatly improved the sensitivity of conserved residue assignment for each human GPCR.

To link receptor evolution to its function, we identified residues that are conserved within the functionally equivalent orthologs for each aminergic receptor. For the residues that play a role in common receptor functions, we expect both clades to retain the amino acid residues with similar physicochemical properties. On the other hand, in the positions that serve receptor-specific functions, in our case the coupling selectivity, we expect to see differential conservation (Fig 1A). Therefore, we grouped receptors based on their known coupling profiles for 11 different G proteins (Fig 1B). We termed these groups as couplers (e.g., $G_s$ coupler receptors) and non-couplers, and performed a two-step enrichment method (Fig 1B) to distinguish specifically conserved residues in couplers from non-couplers. Initially, we used a specific approach to identify evident differentially conserved amino acid residues with high confidence. With the specific approach, residues were labeled as specifically conserved when there was a variation between the coupler and non-coupler receptors but not within coupler receptors (Fig 1B, red and blue arrows). This approach depends solely on the coupling profile datasets (Inoue et al, 2019; Avet et al, 2022) and thus, they may contain false-positive couplings. To tolerate the insensitivity introduced by potential false-positive couplings, we developed and used a sensitive approach enabling to obtain a more complete set of residues for each G protein subtype by allowing minor variations within the coupler receptors. With this method, we used a single comprehensive multiple sequence alignment (MSA) that combined all coupler receptors and their orthologs (Fig 1B, orange arrows), allowing minor variations within a group. We did not apply a sensitive approach to $G_{12}$ and $G_{13}$ because the low number of coupler receptors would likely cause a high number of false positives. Finally, we compared each aminergic receptor and identified positions that were conserved across all aminergic receptors (consensus) to link the specifically conserved residues to the general mechanism of receptor activation. In total, we identified 53 specifically conserved and 22 consensus residues. The distribution of the specific residues for each G protein is presented in Fig 1C.

We aimed to validate the functional impact of potentially deleterious variants that we observe within non-coupler receptors. Thus, we used a dataset (Jones et al, 2020) containing $G_s$ activity scores at EC100 for each possible mutation of ADRB2. 31 residues were identified for $G_s$ and the activity scores of non-coupler variants were plotted (Fig 1D). Non-coupler variants that we identified predominantly decrease $G_s$ activity when compared to the average activity of tolerant substitutions. Under normal conditions, the decrease in $G_s$ coupling can be attributed to various reasons including misfolding and decreased cell surface expression. However, the substitutions we proposed are not likely to disrupt general receptor functions because the substituting amino acids are indeed found and tolerated in non-coupler receptors (Fig 1E) having very high sequence and functional similarity. Additional to the $G_s$ coupling dataset, Kim et al (2020) mutated two of the residues we identified for $G_q$ coupling (8x47 and 6x37) to alanine and showed a decrease in $G_q$ activity compared to WT 5HT2A receptor which validates that variations at specifically conserved positions are not well-tolerated.

The experiments we mentioned show that non-coupler variants cause loss of function in receptors. However, losing the coupling function may not be associated with G protein–coupling selectivity. For an amino acid to be involved in G protein–coupling selectivity, it should govern functional G protein–specific roles. These roles can be recognition of G protein, ligand binding and/or establishing allosteric receptor conformations that may favor (or disfavor) the engagement with certain G protein subtypes. Hence, we manually assigned each residue into functional clusters such as coupling interface and ligand binding. For example, our method identified positions that are at the G protein–coupling interface such as 8x47 (Maeda et al, 2019; Kim et al, 2020; Zhuang et al, 2021b) and 6x36 (Rasmussen et al, 2011; Yang et al, 2020; Xiao et al, 2021) with no structural information taken into account. The residues that are in the coupling interface are in line with the model that Flock et al (2017) proposed and are likely important for proper G protein recognition. However, for the residues that we could not directly assign a role in G protein–coupling activity, we hypothesized that they should be a part of a network controlling the signal transduction from ligand-binding pocket to the G protein–coupling interface and establish the required selective structural conformations. To test this hypothesis, we explored the residue-level contact changes upon coupling to a G protein. We used an algorithm that is called Residue–Residue Contact Score (RRCS) which has been proposed to identify the common activation mechanism in class-A GPCRs (Zhou et al, 2019). We calculated ΔRRCS for each interacting residue pair by subtracting the contact scores of the active structure from the inactive structure. All the

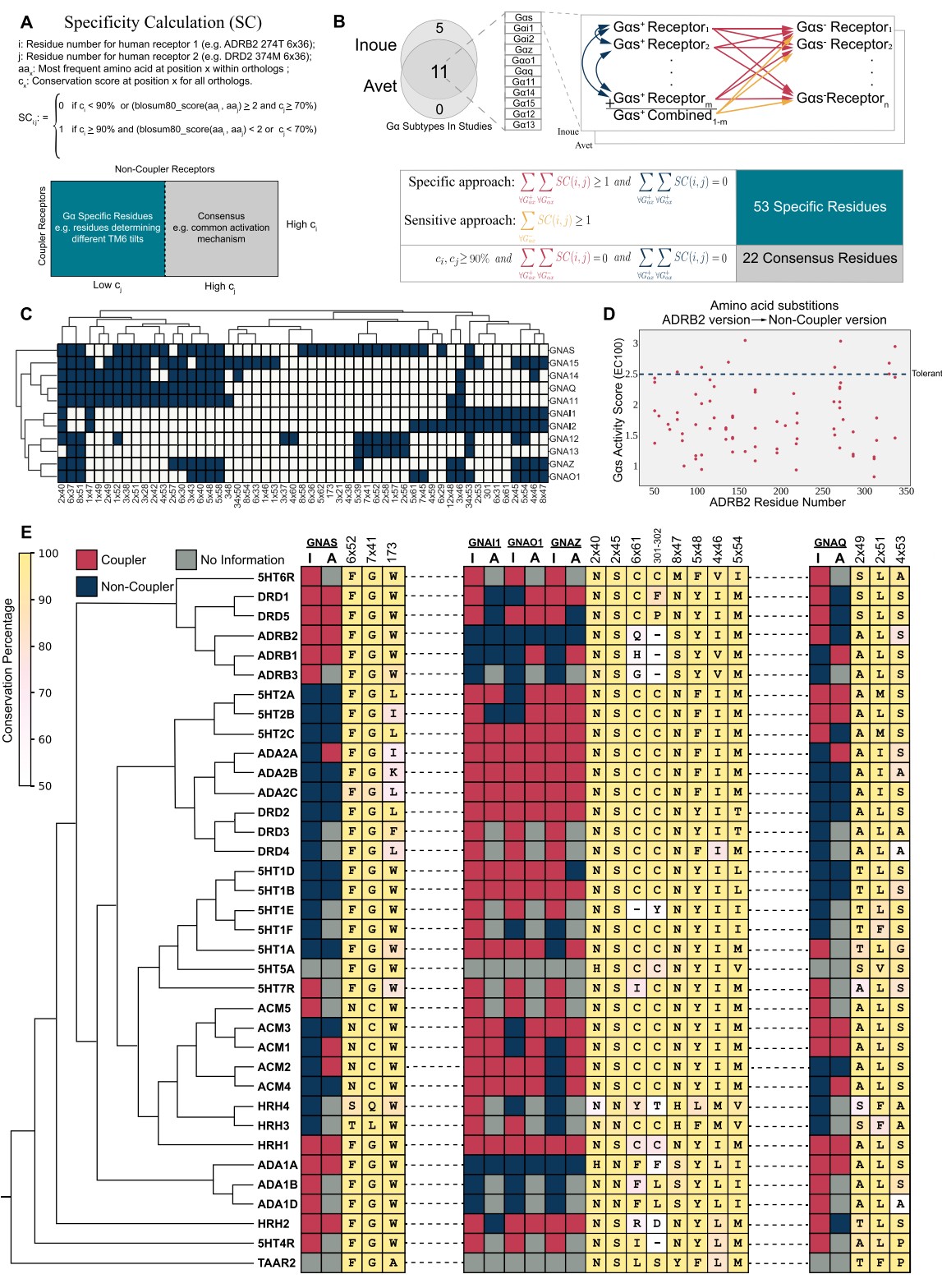

**Figure 1. Selectivity-determining residues for each Gα subtype.**

**(A)** The formula for specific residue identification. **(B)** The schema describes the comparisons between paralogous human receptors to find the specifically conserved residues for each Gα. Arrows represent a single comparison. **(C)** The distribution of specifically conserved residues for each Gα subtype and hierarchical clustering of them (complete linkage). **(D)** Possible variants of G$_s$ specific residues that are observed in non-coupler receptors are compared with the WT activity score. **(E)** Maximum-likelihood phylogenetic tree of aminergic receptors including coupling profiles, conservation information of selected specifically conserved residues (I, Inoue; A, Avet), The background color scale for each consensus amino acid correlates with their conservation (identity).

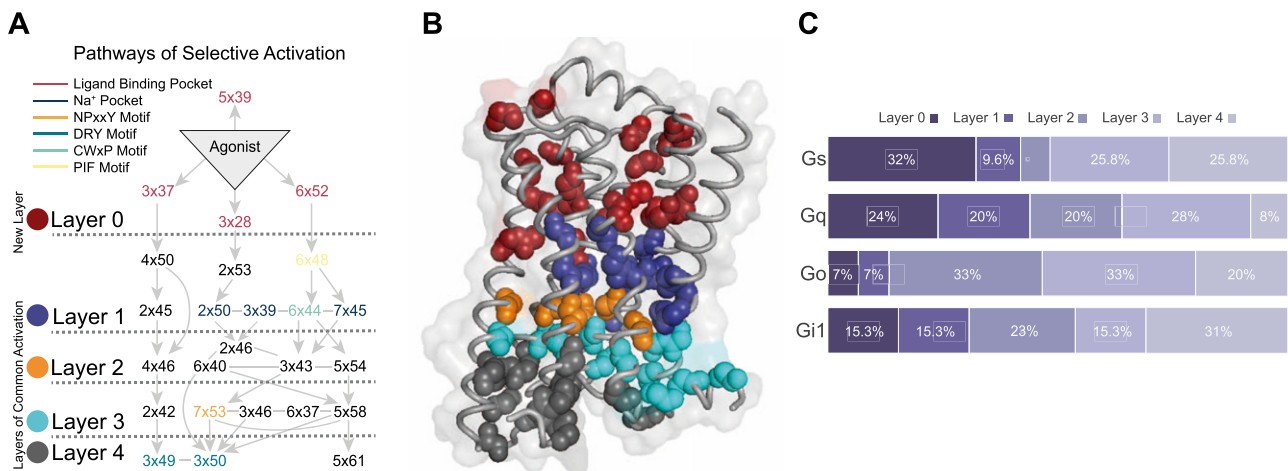

**Figure 2. Structural analysis of molecular pathways that are observed upon coupling with a heteromeric G protein complex.**
**(A)** The most common molecular signal transduction pathways from ligand-binding pocket to G protein–coupling interface. The arrows represent a contact change upon coupling to a G protein. The network is summarized and divided into different layers based on their functional relevance. **(B)** Projection of main chains of specifically conserved and consensus residues in different layers of activation on inactive ADRB2 structure (PDB ID 2RH1). **(C)** The distribution of specifically conserved residues for each analyzed Gα subtype.

active structures we used contained a heteromeric G protein machinery coupled to a receptor. We filtered out residue pairs with |ΔRRCS| ≤ 0.2 and only kept residues that are in our pool of conserved residues (75 residues in total). We analyzed the structures of eight different receptors with four different G proteins (see the Materials and Methods section). The structures we used were experimentally characterized except for one state of a single receptor. As we aimed to use the 10 active-state $G_s$-coupled structures of DRD1, which lacks an experimental inactive structure, we used a model inactive DRD1 structure (Pándy-Szekeres et al, 2018) retrieved from GPCRdb (Kooistra et al, 2021).

In total, we analyzed 41 pairs of active and inactive structures and identified ΔRRCS values of activation networks. We analyzed each network and detected edges (increase or decrease in contact score) observed at least 36 times regardless of the sign of ΔRRCS value to build a network that would represent all 41 networks. By using this network, we identified the most frequently used signal transduction paths (Fig 2A), connecting ligand-binding pocket to G protein–coupling interface and creating a basis for the routes that can induce coupling selectivity. We divided the receptor into five layers based on the sequential nature of interactions and illustrated the direction of signal transduction between layers. Additional to the four layers (1–4) that were previously proposed in the common activation mechanism (Zhou et al, 2019), we defined "Layer 0" which corresponds to the ligand-binding site. Though most of the signaling paths pass through important motifs such as $Na^+$ binding pocket and PIF (Katritch et al, 2014), it is remarkable that the novel path starting with a 3x37 does not require the involvement of any of these important motifs. Within the identified network, the signal is transmitted from the ligand-binding pocket to the G protein interface by using mainly TM2, TM3, and TM4. We projected all the residues onto an inactive structure of ADRB2 based on the layers they belong to (Fig 2B) to provide an insight into the locations of different layers.

To determine the contribution of each layer for $G_s$, $G_{i1}$, $G_o$, and $G_q$, we calculated the distribution of specific residues to different

layers (Fig 2C). Layer 0 and Layer 1 are more involved in the coupling for $G_s$ and $G_q$ relative to $G_{i1}$ and $G_o$. For $G_o$, 86% of the coupling-related residues are positioned in the layers (2, 3, and 4) closer to the G protein–binding site. Differences in these distributions indicate mechanistic differences between distinct coupling events.

To detect if the specifically conserved residues have differential roles in G protein coupling–related mechanisms, we grouped ΔRRCSs (contact changes upon coupling to a G protein) for the receptors coupled to the same G protein and compared them with the rest by using two-sample $t$ test. This approach yielded interaction changes (ΔΔRRCS) within the receptors that are significantly different ($P < 0.01$) and specific for $G_s$, $G_{i1}$, $G_o$, and $G_q$. Significant contact changes occurring between 75 conserved residues were used to construct G protein–specific activation mechanisms. The constructed networks (Fig 3B–E) support our evolution-driven hypothesis and demonstrate that specifically conserved residues indeed have differential mechanistic roles in G protein coupling. In parallel to Fig 2C, networks for $G_s$ and $G_q$ contained ligand contacting residues (Fig 3A and E), whereas networks for $G_{i1}$ and $G_o$ do not. This can indicate that ligand binding could be more important for $G_s$ and $G_q$ coupled receptors. Although, $G_{i1}$ and $G_o$ belong to the same subfamily and they share eight of the specifically conserved residues (47% of the specifically conserved residues for $G_o$ and 62% for $G_{i1}$) of G proteins their networks are totally different from each other. Moreover, even when we grouped the receptors coupled to Gi together, no significant difference in contact scores having a $P$-value less than 0.01 was observed for the shared specifically conserved residues (Fig 1C). This suggests that receptors coupling to Gi may not necessarily share a common activation mechanism. Therefore, these differences in activation networks could be one of the factors determining selectivity between $G_{i1}$ and $G_o$ coupled receptors.

Even though residues specifically conserved for the receptors sharing similar coupling profiles are part of G protein–specific activation networks, it is still not clear that these contact changes

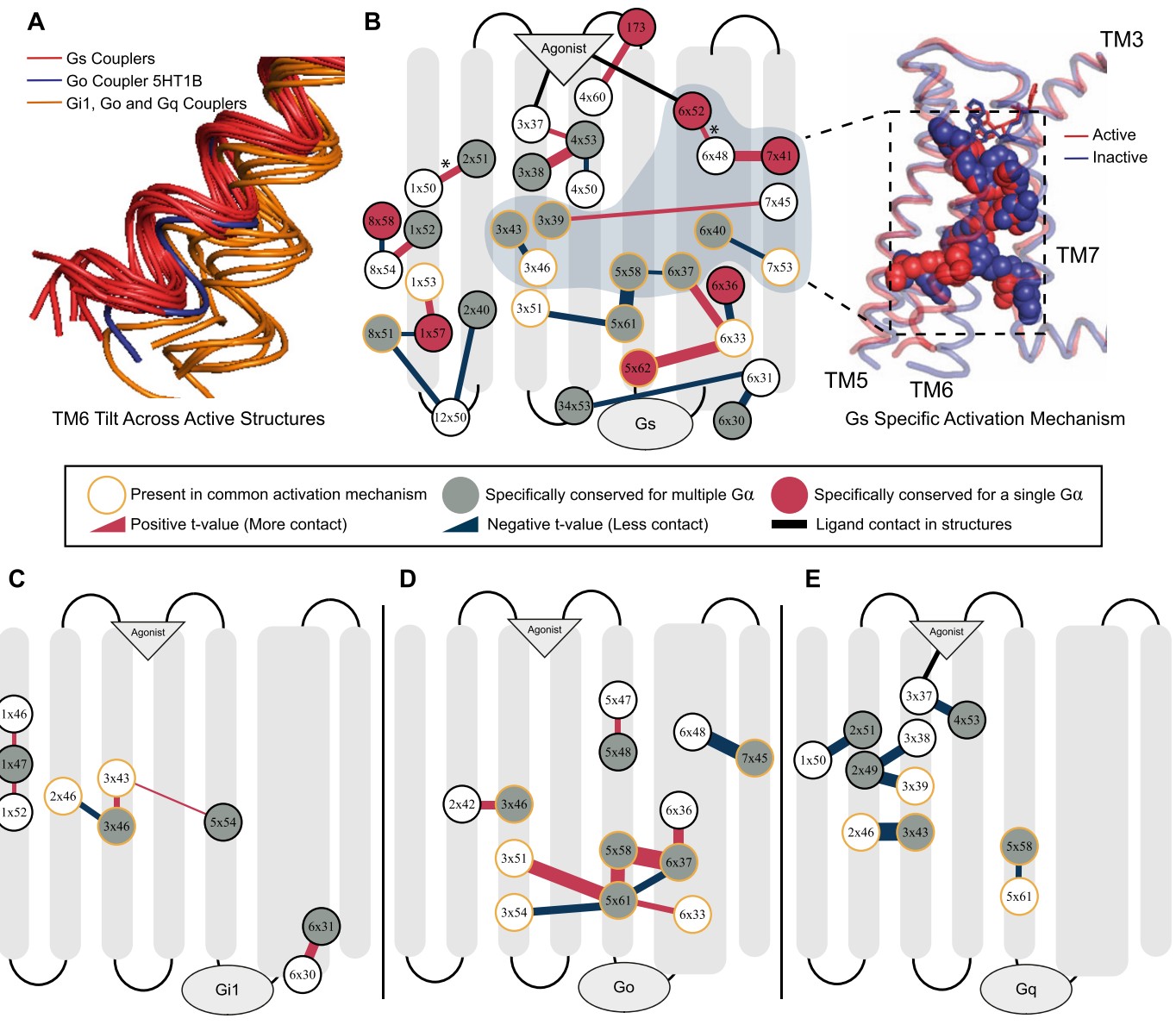

**Figure 3. Specific activation networks for G$_s$, G$_{i1}$, G$_o$ and G$_q$.**
**(A)** TM6 tilt comparison between the active receptors we used. Red: G$_s$ couplers, Orange: G$_o$, G$_{i1}$, and G$_q$, Blue: 5HT1B G$_o$ coupler as an exception. **(B)** Interactions within the receptor that are specific (*P* < 0.01) to G$_s$. Red: increasing contact, blue: decreasing contact, orange circle: present in common activation mechanism, red fill: uniquely identified specific residue for G$_s$, grey fill: G$_\alpha$ specific residue. The width of the lines correlates with statistical significance. A group of residues that possibly facilitate in TM6 movement for G$_s$ coupling was shown on inactive (blue) and active (red) structures. **(C, D, E)** Specific interaction networks for G$_{i1}$, G$_o$, and G$_q$. *P* < 0.1 is used for G$_{i1}$. *: This interaction is identified only if 5HT1B is neglected from the comparison because of its larger TM6 movement.

are the basis for selective coupling, or they arise due to the physical interaction with a G protein itself. To show that these networks can determine selectivity we further analyzed the activation network for G$_s$-coupled receptors. Previously, it was shown that receptors coupled to G$_s$ achieve a larger TM6 tilt (Rose et al, 2014; Van Eps et al, 2018) than the receptors coupled to other G proteins. Superimposition of the active structures that we used in our analysis (Fig 3A) is also in line with the previous findings. We hypothesized that if differential TM6 movement is a determinant for G$_s$, the network we identified can modulate this structural difference. Furthermore, the requirement for a larger TM6 movement can be the reason why G$_s$

specific activation mechanism is more complex than the rest (Fig 3B–E). An exception to this is the TM6 position of 5HT1B (García-Nafría et al, 2018) that is coupled to G$_o$ (Fig 3A, blue structure) because it achieved a slightly larger tilt. Thus, we performed an additional statistical test to reveal possible interactions that can promote larger TM6 movement by excluding the samples for 5HT1B and revealed the 6x52–6x48 interaction indicating the role of 6x48 in differential TM6 movement in G$_s$-coupled receptors (*P* = 0.0023).

We projected a part of G$_s$ specific activation network which we predicted to be associated with the differential TM6 movement onto experimentally resolved active (red, 3SN6) and inactive (blue,

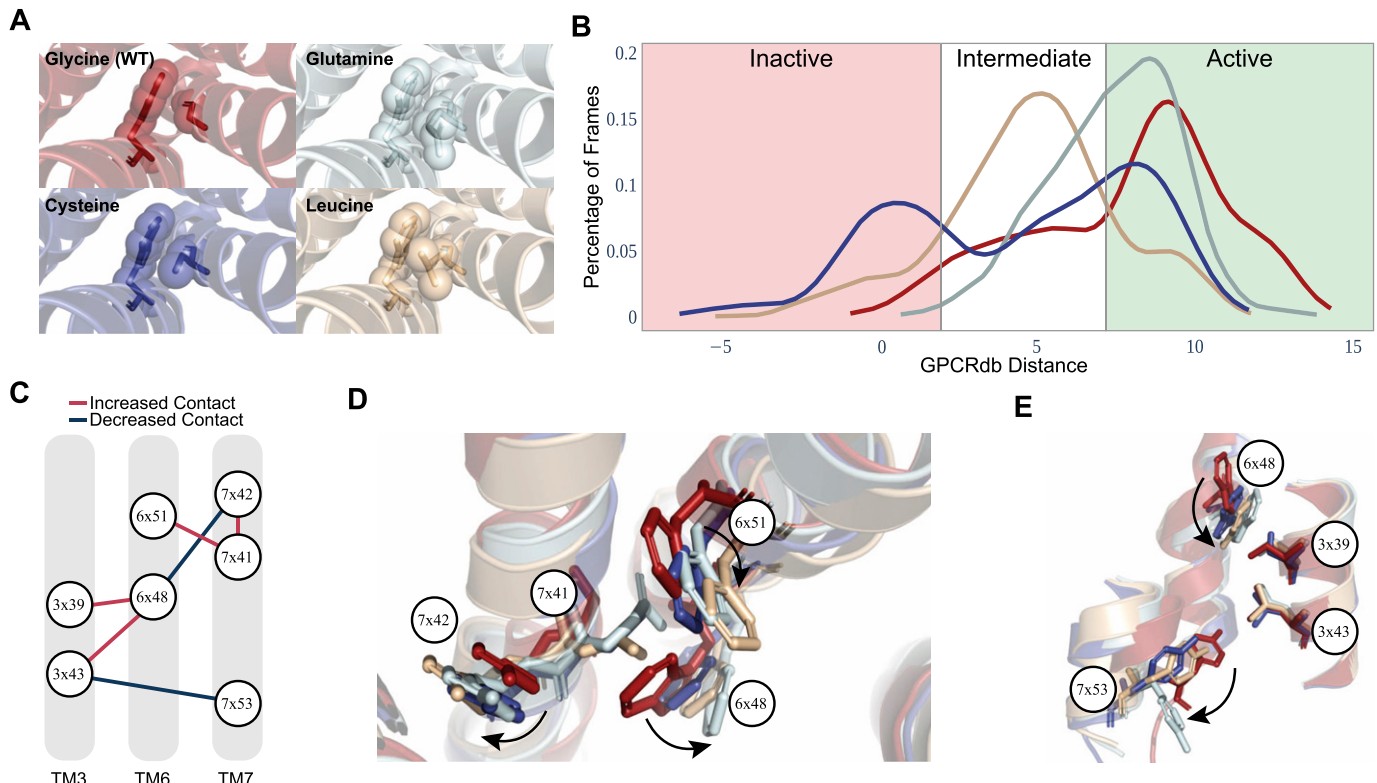

**Figure 4. Analysis of molecular dynamics simulations reveals functional importance of glycine at 7x41.**
**(A)** Four different MD simulation systems were shown in their initial conformation. **(B)** For each simulation, distribution of frames with respect to their state of activation was shown, distance in Angstrom. **(C)** The common pathway representing the impact of the mutations at 7x41. **(D, E)** The common pathway was represented on average structures that were obtained in all MD trajectories for every mutation and WT. The movements of residues were represented with arrows.

2RH1) ADRB2 structures (Fig 3B). More specifically, we hypothesized that the network containing 6x52 and 7x41 triggers this structural difference because interactions at the upper layers are more likely to be leading a structural change. In agreement with our hypothesis, deep mutational scanning of ADRB2 (Jones et al, 2020), has revealed that 7x41 is the second and 6x48 is the fourth most intolerant residue to any mutations and, to our knowledge, no previous study has investigated the functional role of 7x41 until now. It is expected that a position that is crucial for G$_s$ coupling to be one of the most intolerant residues for a receptor primarily coupled to G$_s$.

To validate our methodology and further understand the mechanistic insight of the relevance of the core transmembrane region in G protein coupling, we studied the glycine at position 7x41 as a test case and performed molecular dynamics (MD) simulations. We applied three different mutations, G315C, G315Q, and G315L, on monomeric active and inactive-state ADRB2 (Fig 4A). We particularly selected variants observed in acetylcholine and histamine receptors (Fig 1E) to validate our hypothesis that variants in non-coupler aminergic receptors at the same position are inactivating. We used two main metrics to assess the molecular impact of these three mutations. First, the comparison active/inactive states based on GPCRdb distances (see the Materials and Methods section) revealed that the WT receptor preserve the active state more than the variants (Fig 4B) and leucine residue was the most inactivating mutation. The significant inactivation through the integration of

leucine mutation is parallel to pre-existing experiments (Arakawa et al, 2011; Jones et al, 2020). Then, to identify the molecular changes in absence of glycine, we evaluated the significant contact differences (ΔRRCS) between WT and mutated MD simulation trajectories.

To examine the entire trajectory, we selected 11 frames from each simulation with 50-ns time intervals (in total 500 ns) for each replicate. Thus, we compared RRCS of 77 mutated and 77 WT frames for active-state simulations, whereas we compared 22 mutated and 22 WT frames for inactive-state simulations by using two-sided $t$ test. For each mutation and activation state, we identified significant contact changes ($P < 0.01$) and intersected common changes that we observed for all of the mutated systems. As a result, we identified 135 residue pairs for active and 83 residue pairs for inactive simulations. When we projected these residue pairs (135 residue pairs) as a contact network, we identified a conserved and highly affected pathway (Fig 4C) connecting ligand-binding pocket to NPxxY motif which showed changes towards the inactivation of the receptor. Then, we projected the identified molecular pathway onto average cluster structures that were produced by using the trajectories from all seven replicates (35,000 frames in total) for each mutation (Fig 4D and E). MD results suggested a pathway (Fig 4C) that explains the importance of G315: An increased bulkiness of the amino acid at 7x41 (by non-glycine amino acids) leads to increased contact with 7x42 and 6x51, whereas 7x41 physically impairs the interaction between 6x48 and 7x42. When 6x48 loses its contact

with 7x42 (Fig 4D), it increases its contact residues at TM3 3x43 and 3x39 (Fig 4E). Increased interactions between TM6 and TM3 loosen TM3-TM7 packing which is an important initiator of the TM6 tilt in class-A GPCRs (Zhou et al, 2019). In addition, it loosens the contacts between TM6 and TM7 through 6x48-7x42, 6x44-7x49, and 6x52-7x45, which explains the increased distance between 7x53 and 3x43 (Fig 4E). Moreover, the simulations of cysteine and leucine variants exhibited an increased contact between 3x43 and 6x40 ($P < 0.01$) inhibiting the receptor activation through restricting outward TM6 movement. When we evaluated the inactive trajectories, we observed similar contact changes between 6x48, 6x51, 7x41, and 7x42 ($P < 0.01$) proving that the simulation results are not biased toward active-state simulations. Thus, analysis of MD trajectories suggests that glycine at 7x41 plays an important role in receptor activation, and it is likely to control selectivity for $G_s$ coupling by promoting a larger tilt of TM6 which we observe almost exclusively in $G_s$-coupled receptors. However, the roles of $G^{7x41}$ and differential TM6 position in determining $G_s$ coupling selectivity requires experimental validation.

## Discussion

By integrating our findings and current literature we propose a G protein selectivity model involving a series of modules. As pilots turn on switches in a pre-determined order before the takeoff, GPCRs must turn on their molecular switches for a specific type of G protein coupling to occur. If pilots fail to turn on all the switches properly because of an error, there will be no permission for them to depart. Similarly, all molecular switches must be turned on for receptors to engage with a G protein and induce downstream signaling pathways. For these reasons, we named our model "sequential switches of activation" (Fig 5). We propose the existence of three main switches within a GPCR structure. The first switch checks for binding of the proper agonist which induces conformational changes in the lower layers of the receptors. If an agonist makes the proper contacts with the receptor the first switch turns on. Then as a next step, receptors should be activated through G protein selective activation mechanisms which include multiple micro-switches to turn off the second main switch. Micro-switches represent the arrangement of inner contacts that are specific for G protein subtypes. When inner contacts are established properly the second switch turns on as well. As a third and last checkpoint, receptors should contain the set of residues that can recognize the ridges on G proteins according to the "key and lock" model that Flock et al (2017) suggested. When required contact between G protein and receptor is established, the third switch turns on and the receptor is successfully coupled by a subtype of G proteins. Mutations inducing constitutional activity can be considered as a "short circuit" because they can bypass switches. On the other hand, mutations that halt the receptor's ability to turn on a particular switch can prevent coupling. It is important to note that our model is inclusive of and complementary to the model Flock et al (2017) suggested. The combination of these two models gives us a more complete perspective on receptor-level determinants of coupling selectivity.

In our study, we used a novel phylogenetic approach to identify residues that are conserved among groups of receptors coupling to

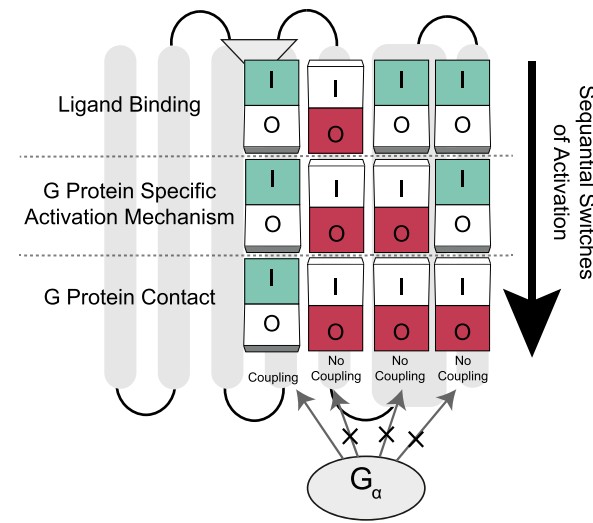

**Figure 5. Sequential switches of activation model for G protein selectivity.** The model describes that all switches in different layers of receptors must be turned off for receptor activation and coupling of the G protein. If switches at upper layers are halted due to a mutation, the following switches become turned off which inhibits G protein coupling eventually.

a particular G protein. We identified the largest possible set of residues (Fig 1C) by combining sensitive and specific approaches together. Because of our greedy approach, whereas some positions could determine coupling selectivity, others may be "passenger" positions that may modulate core receptor functions. Moreover, the positions we identified are the ones that are shared among all aminergic receptors and lack receptor-level variations. Previous studies on chimeric GPCRs (Wess, 1998, 2021; Wong, 2003) point out the importance of ICL3 in determining coupling selectivity. Although we identified residues that contact with G proteins, our analyses did not reveal any possible determinants at ICL3. This indicates that the determinants at ICL3 are not shared between aminergic receptors and rather be specific to individual receptors. Alternatively, in nature, there may not be a solution for G protein–coupling selectivity determination with ICL3 only. Experimentally constructed chimeric receptor activation should be handled with caution because they cannot be evaluated as a part of receptor evolution. Thus, to identify all selectivity-determining positions, each receptor should be analyzed individually.

Although our study does not include any direct experimental evidence that coupler or non-coupler variants alter coupling selectivity, it provides sufficient evidence to support the existence of receptor-wide selectivity determinants not only at the G protein–coupling site but throughout receptors including the ligand-binding site. A recent analysis (Seo et al, 2021) also shows that serotonin and dopamine receptors contain positions co-evolving with the positions on G proteins they are coupled to. Supporting our conclusions, selectivity-determining positions were shown dispersed throughout the receptor. In our study, we used $G_s$ coupling data from deep mutational scanning of ADRB2 performed by Jones et al to show that non-coupler variants cause loss of function (Fig 1D) (Jones et al, 2020), their roles in determining coupling selectivity should be clarified further. For that purpose, we used the RRCS

algorithm and revealed the involvement of specifically conserved residues in G protein–specific activation mechanisms (Fig 3B–D) which suggests their role in determining coupling selectivity. We should note that because of the scarcity of $G_q$-, $G_o$-, and $G_{i1}$-coupled structures, the networks we provided could be modified in the future as the number of G protein–coupled experimental structures increases. As a third layer of evidence, we identified the role of a previously uncharacterized $G^{7x41}$ (Jones et al, 2020) for ADRB2 and $G_s$-coupled receptors through MD simulations (Fig 4C). Although we cannot rule out the potential effect of $G^{7x41}$ in non-$G_s$ activation, we can conclude that it has a critical importance for determining $G_s$ coupling selectivity. The fact that $G^{7x41}$ is dispensable for Gi couplers suggests that it may not be as critical for those GPCRs and Gi activation. To summarize, multiple layers of evidence suggest that G protein selectivity determinants for aminergic receptors are likely distributed receptor-wide.

The conclusions of this study are limited aminergic receptors only because there has been no supporting evidence for a common selective mechanism that might present for all class-A GPCRs. Therefore, it is necessary to handle each GPCR subfamily separately to identify subfamily-specific selectivity determinants. With such an effort, it may be possible to discover commonalities and differences between different subfamilies of GPCRs. Although different sub-families of receptors couple with a G protein by having similar structural conformations, underlying mechanisms for achieving a conformation might vary. As the number of solved G protein-coupled receptor structures increases in the protein data bank, it is inevitable that new selectivity determinants and similar mechanisms will be discovered in near future.

# Materials and Methods

### Sequence selection

Sequence selection is the very first step of this study. We used the BLAST+ (Camacho et al, 2009) algorithm to obtain homologous protein sequences from other organisms. We blasted a human target protein to find its homologs. The UniProt (UniProt Consortium, 2019) database is used as a source for the sequences. We re-trieved all the sequences until the third human protein from the blast output.

### MSA #1

After sequence selection, the next step is performing MSA for obtained sequences. For this purpose, we used MAFFT (Katoh & Standley, 2013) "einsi" option which allows for large gaps. This option allows us to align multiple homologous regions of different receptors.

### Maximum likelihood (ML) tree #1

The MSA was used to produce a maximum-likelihood (ML) tree. ML trees helped us find relationships between different proteins. ML Tree 1 was used to identify the clade which contains our protein of interest. For ML tree construction we use the IQ-TREE version 2.0.6

(Minh et al, 2020) We used 1,000 ultrafast bootstraps and JTT+I+G4+F substitution model. IQ-TREE is used at this step for mainly its high speed in bootstrapping.

### Obtaining gene clade

For making modifications to the ML trees we use a Python-based tool ETE3 (Huerta-Cepas et al, 2016). To analyze a tree, we first need to root it properly. We chose the third human protein from our BLAST results, as an outgroup. Then, we traversed from our target human leaf node to root until we reached a clade containing another human protein. After each move, we analyzed the species content of the clades we are observing. When a clade contained species that were not observed in previous moves, we included all of the leaf nodes in our analysis. On the other hand, when a clade contains a previously observed species, we exclude that clade from our analysis because seeing a species at lower phylogenetic levels is an indication of a differential gene loss event. We continued with the remaining sequences and produced a MSA with them.

### MSA trimming

MSA trimming is needed to remove some of the noise from the alignment and it speeds up tree reconstruction. MSA trimming removes positions that are misleading for tree production. For example, positions having too many gaps can be removed from the alignment. We used trimAl (Capella-Gutiérrez et al, 2009) with automated1 option which is stated to be the best option for constructing maximum-likelihood trees.

### Maximum likelihood tree #2

ML tree 2 was used to identify the paralogous sequences that we have in our analysis. For ML tree construction, we used the RaxML-NG version 0.9.0 (Kozlov et al, 2019) –search option with JTT+I+G4+F substitution model.

### Paralog trimming

Paralog trimming is a key part of our approach. After gene dupli-cation, one of the paralogous clades tends to diverge more than the other. Unless the diverged clade is removed from our analyses (MSA), it might introduce false divergence signals in conservation calculation. For this reason, we need to exclude diverged paralogs from our analyses. We used the second ML tree for the detection of the diverged paralogs.

We first calculated the global alignment scores (BLOSUM62 is used) of every sequence on the ML tree 2 with respect to our human target sequence. We assessed each internode having two child clades based on the number of leaf nodes and species they contain. When two child clades contained at least one identical species, we looked for a significant divergence between the clades in terms of global alignment scores to label one clade as paralogous. Also, we need those clades to be evolutionarily comparable, thus we compared the taxonomic level of the organisms between two clades. If the clades are comparable with each other, we applied two-sample $t$ test for by using the global alignment scores. If one

clade has significantly lower similarity scores ($P \leq 0.1$) that clade is labeled as a diverged paralogous clade. We applied the same approach for detecting the taxonomic level of the organisms and common lineage numbers with *Homo sapiens* were used this time ($P \leq 0.1$). If the clades are evolutionarily comparable and one clade had a significantly lower global alignment score, all of the sequences belonging to that clade were eliminated.

When two of the clades contained less than three sequences each, it was hard to obtain significance. Therefore, for those cases, we compared the average global alignment scores and eliminated the clade with a lower average. For the remaining situations, we do not remove any of the clades.

## Conservation calculation

After obtaining orthologs we used them to calculate the conservation scores for each receptor.

The conservation percentage for a certain residue is calculated as follows:

(1) Find the most frequent amino acid for a certain position in the MSA.
(2) After finding the most frequent amino acid, we compared it with other alternatives in that position. When comparing amino acids, we calculated BLOSUM80 score for each of them. If the BLOSUM80 score is higher than 2 we accept it as an "allowed" substitution because it means that these amino acids replace each other frequently and have similar properties.
(3) The gaps are not included while calculating the conservation percentage.
(4) If gaps are more than 50%, we categorized that position as a gap.
(5) The conservation score is equal to the number of most frequently observed and "allowed" amino acids over the number of all non-gap positions

## Specificity calculation

For a position to be specific or consensus the criteria are the following:

(1) First, we need one alignment of two proteins with their orthologs. Then we split the alignment into two alignments with the same length.
(2) We label a position as consensus when both alignments are conserved more than the consensus threshold (90%) at that particular position and the most frequent amino acids are similar (BLOSUM80 score is more than 1) to each other.
(3) We calculated conservation percentages for each alignment. There are two different scenarios in this case. The first one is when the most frequent amino acids of the two of the alignments are not similar (BLOSUM80 score is lower than 2) to each other. If this is the case and the conservation percentage for any alignment is above the specificity threshold (90%) we label that position as specifically conserved for that alignment. The second case is where the most frequently observed amino acids are similar to each other. In this case, for a position to be specific for one alignment first it should satisfy the specificity

threshold and secondly, the conservation percentage of the other alignment should be lower than our lower threshold (70%).

For the steps above, we choose 90% for both specificity and consensus thresholds. 70% is selected for the lower specificity threshold.

## Enrichment of specifically conserved residues

We identified specifically conserved residues with two different approaches:

### Specific approach

(1) We divided receptors into two couplers versus non-couplers. Let us assume that we have n number of couplers and m number of non-couplers.
(2) We compare coupler receptors with non-couplers in a pairwise manner. In these comparisons, we count the number of being specific for every residue. In total there are n times m comparisons. We divide the obtained counts by the total number of comparisons to get the frequency of a residue being specific for the couplers' group.
(3) To examine if a residue is generally variable or specific to the coupling event, we compared couplers with themselves. We applied STEP 2 for couplers–couplers comparison as well. This time, we have n × (n − 1) comparisons in total. We again calculated the frequencies accordingly.
(4) For the specific approach, we do not allow any inside variation and this makes the result of STEP 3 zero. On the other hand, for a residue to be labeled as specific, we expect STEP 2 more than zero. When these two conditions are satisfied, we label that residue as specifically conserved

### Sensitive approach

(1) We built a comprehensive MSA for the coupler receptors and their orthologs.
(2) We compared this alignment with non-coupler receptor's MSAs similarly to STEP 2 of the Specific Approach.
(3) We added newly discovered positions to our analysis as specifically conserved.

### Building the maximum-likelihood phylogenetic tree for aminergic receptors
(1) We blasted (Camacho et al, 2009) aminergic receptors and obtained the first 50 sequences to generate a fasta file.
(2) From that fasta file we selected representative sequences by using cd-hit default options.
(3) MAFFT (Katoh & Standley, 2013) einsi algorithm was used to align representative sequences.
(4) IQ-TREE version 2.0.5 (Minh et al, 2020) was used to create the phylogenetic tree with options: -m JTT+G+I+F -b 100 –tbe

## RRCS and network analysis

We calculated the RRCS score for 20 active (ADRB2: 3SN6, 7DHI; DRD1: 7CKW, 7CKX, 7CKZ, 7CKY, 7CRH, 7JV5, 7JVP, 7JVQ, 7LJC, 7LJD; DRD2: 6VMS,

7JVR; DRD3: 7CMU, 7CMV; 5HT1B: 6G79; ACM2: 6OIK; 5HT2A: 6WHA; HRH1: 7DFL) (Rasmussen et al, 2011; García-Nafría et al, 2018; Maeda et al, 2019; Kim et al, 2020; Xia et al, 2021; Xiao et al, 2021; Xu et al, 2021; Yang et al, 2020; Yin et al, 2020; Zhuang et al, 2021a, 2021b) and 24 inactive structures (ADRB2: 2RH1, 6PS2, 6PS3, 5D5A; DRD1: GPCRdb inactive model; DRD2: 6CM4, 6LUQ, 7DFP; DRD3: 3PBL; 5HT1B: 4IAQ, 4IAR, 5V54, 7C61; ACM2: 3UON, 5YC8, 5ZK3, 5ZKB, 5ZKC; 5HT2A: 6A93, 6A94, 6WH4, 6WGT; HRH1: 3RZE) (Cherezov et al, 2007; Chien et al, 2010; Shimamura et al, 2011; Haga et al, 2012; Wang et al, 2013, 2018b; Huang et al, 2016; Suno et al, 2018; Yin et al, 2018; Ishchenko et al, 2019; Kimura et al, 2019; Fan et al, 2020; Im et al, 2020; Kim et al, 2020; Miyagi et al, 2020). For each receptor, we subtracted inactive RRCS from active RRCS to obtain ΔRRCS values for each residue pairs. We wrote a custom python code to obtain files with ΔRRCS scores. We combined all of the networks that contain information about the contact changes upon activation to produce the most common molecular signal transduction pathways (Fig S1). For the details of the RRCS algorithm, please read the corresponding article (Zhou et al, 2019).

### Identification of G protein–specific activation networks

After obtaining ΔRRCS networks for each active–inactive structure pair we grouped ΔRRCS values based on the G protein subtype coupling the receptors. Then we compared ΔRRCS values of individual groups (e.g., $G_s$: ADRB2 and DRD1) with the rest of the groups (e.g., Non-$G_s$: DRD2, DRD3, 5HT1B, ACM2, 5HT2A, HRH1) by using two-sample $t$ test. Whereas $P \leq 0.01$ is used for $G_s$, $G_q$, and $G_o$, $P \leq 0.1$ is used for $G_{i1}$. We obtained significant contact changes upon coupling to a particular G protein.

### Molecular dynamics simulations

We downloaded inactive and active structures of Beta2 Adrenergic receptor ($\beta_2$AR) from PDB (PDB ID: 4GBR, and 3SN6, respectively) (Rasmussen et al, 2011; Zou et al, 2012). Three thermostabilizing mutations, T96M[2x66], T98M[23x49], and E187N[ECL2,] were mutated back to the WT in both sequences. Because the used inactive structure of the $\beta_2$AR has a short ICL3 that links the TM5 and TM6, we did not introduce additional residues to the ICL3 and used the crystal structure as it is. However, the active structure of the $\beta_2$AR lacks ICL3, and we modeled a short loop with GalaxyLoop code (Park et al, 2014). We inserted FHVSKF between ARG239 and CYS265. We introduced three changes at the 315[7x41] position, and one WT and obtained three mutants (namely; G315C, G315L, and G315Q). We used PyMOL to place mutations (PyMOL Molecular Graphics System, Version 2.1.0.). Orientations of proteins in biological membranes were calculated with OPM server (Lomize et al, 2012) and We used CHARMM-GUI web server (Jo et al, 2008; Wu et al, 2014; Lee et al, 2016) to create input files for the MD simulations for Gromacs. Because inactive and active structures start with ASP29[1x28] and GLU30[1x30] and end with LEU342[Cterm] and CYS341[8x59], respectively, we introduced acetylated N-terminus and methylamidated C terminus to the N- and C-terminal ends. Two disulfide bridges between CYS106[3x25]-CYS191[ECL2] and CYS184[ECL2]-CYS190[ECL2] were introduced. Each lipid leaflet contains 92 (1-palmitoyl-2-oleoyl-sn-glycero-3-phosphocholine) POPC biological lipid type (total 192 POPC

molecules in system). Systems were neutralized with 0.15 M NaCl ions (50 Na$^+$ and 55 Cl$^-$ ions in total). We used TIP3P water model for the water molecules (MacKerell et al, 1998) and CHARMM36m force field for the protein, lipids, and ions (Huang et al, 2017). One minimization and six equilibration steps were applied to the systems, before production runs (for the equilibration phases 5, 5, 10, 10, 10, and 10 ns MD simulations were run, in total 50 ns). In equilibration phases, both Berendsen thermostat and barostat were used (Berendsen et al, 1984). In production runs, we applied Noose–Hoover thermostat (Nosé & Klein, 1983; Hoover, 1986) and Parrinello–Rahman barostat (Parrinello & Rahman, 1980). 500 ns production simulations were run with Gromacs v2020 (Abraham et al, 2015) and repeated seven times to increase sampling (in total for each system we simulated 3.5 $\mu$s). 5,000 frames were collected for each run, and for instance for the WT system, we concatenated 35,000 frames to calculate GPCRdb distance distributions (*gmx distance* tool was used for this purpose) and find average structures (Visual Molecular Dynamics code used to find average structure [Humphrey et al, 1996]). To calculate the GPCRdb distance in Class-A GPCR structures, CYS125[3x44]-ILE325[7x52] distance was subtracted from TYR70[2x41]-GLY276[6x38] distance. If the calculated distance is higher than 7.15 Å, lower than 2 Å, and between 2 and 7.15 Å state of the receptors labeled as active, inactive, and intermediate, respectively (Isberg et al, 2015; Shahraki et al, 2021). All figures were generated with PyMOL v2.1.0. To estimate water accessibilities to the internal cavity of the receptors, and sodium ion accessibilities to the ASP79[2x50], we calculated averaged water and sodium ion densities. Time-averaged three-dimensional water and sodium ion density maps were calculated with GROma$\rho$s (Briones et al, 2019).

### Analysis of contact changes within molecular dynamics simulation trajectories

Frames of MD simulation trajectories were selected from 0 to 500 ns with 50 ns gaps for each trajectory and replicate for a mutation. Including the frame at t = 0 ns, for a replicate, we obtained 11 frames to represent the whole trajectory. We have applied the same strategy for all seven active-state replicates and obtained 77 frames for WT and mutated MD trajectories. For each frame, we calculated RRCSs for every residue pair and identified statistically significant ($P < 0.05$) differences between WT and mutated trajectories by applying a two-sided $t$ test. For the inactive simulations, we had only two replicates; therefore, we compared 22 mutated frames with 22 WT frames.

After applying $t$ test, we intersected the significant contact changes we observed for each mutational state to observe the common change due to the absence of glycine. In total, we identified 135 common changes for active-state simulations and 83 common changes for inactive-state simulations. We used Cytoscape (Shannon et al, 2003) to visualize the changes as a network. PyMOL was used to visualize the identified pathway on protein structures.

## Data Availability

The open-source code and supplementary data are available at our GitHub repository: https://github.com/CompGenomeLab/GPCR-coupling-selectivity.

The MD trajectories have been deposited to: https://doi.org/10.5281/zenodo.5763490.

# Supplementary Information

# Acknowledgements

This work is supported by EMBO Installation Grant (to O Adebali 4163) that is funded by TUBITAK. We acknowledge partial support from Turkish Academy of Sciences (TÜBA-GEBIP) and Science Academy, Turkey (BAGEP). The molecular dynamics simulations reported in this paper were fully performed at TUBITAK ULAKBIM, High Performance and Grid Computing Center (TRUBA resources).

## Author Contributions

B Selçuk: conceptualization, software, formal analysis, investigation, visualization, methodology, and writing—original draft.
I Erol: investigation, methodology, and writing—review and editing.
S Durdağı: supervision and writing—review and editing.
O Adebali: conceptualization, resources, supervision, funding acquisition, project administration, and writing—review and editing.

## Conflict of Interest Statement

The authors declare that they have no conflict of interest.

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
