## [Reviewer comments · Life Science Alliance]

Evolutionary association of receptor-wide amino acids with G protein coupling selectivity in aminergic GPCRs

Berkay Selçuk, Ismail Erol, Serdar Durdağı, Ogun Adebali

DOI: 10.26508/lsa.202201439

Corresponding author(s): Dr. Ogun Adebali (Sabanci University)

Review timeline:

Submission Date:	2022-03-08
Editorial Decision:	2022-03-08
Revision Received:	2022-03-16
Editorial Decision:	2022-04-19
Revision Received:	2022-05-04
Editorial Decision:	2022-05-05
Revision Received:	2022-05-12
Accepted:	2022-05-13

Scientific Editor: Eric Sawey

Transaction Report:

Please note that the manuscript was previously reviewed at another journal and the reports were taken into account in inviting a revision for publication at *Life Science Alliance* prior to submission to *Life Science Alliance*.

No Peer Review Process File is available with this article, as the authors have chosen not to make the review process public in this case.

Re: Life Science Alliance manuscript #LSA-2022-01439-T

Ogun Adebali
Sabanci University
Turkey

Dear Dr. Adebali,

Thank you for submitting your manuscript entitled "Receptor-wide Determinants of G Protein Coupling Selectivity in Aminergic GPCRs" to Life Science Alliance. We invite you to submit a revised manuscript with the following revisions:

- Address the Reviewers' comments, excluding Reviewer 3's Comment #3.

Thank you for this interesting contribution to Life Science Alliance. We are looking forward to receiving your revised manuscript.

Sincerely,

B. MANUSCRIPT ORGANIZATION AND FORMATTING:

Re: Life Science Alliance manuscript #LSA-2022-01439-TR

Dr. Ogun Adebali
Sabanci University
Faculty of Engineering and Natural Sciences
Istanbul 34956
Turkey

Dear Dr. Adebali,

Thank you for submitting your revised manuscript entitled "Evolutionary association of receptor-wide amino acids with G protein coupling selectivity in GPCRs" to Life Science Alliance. The manuscript has been seen by the original reviewers whose comments are appended below. While the reviewers continue to be overall positive about the work in terms of its suitability for Life Science Alliance, some important issues remain. Specifically, as pointed out by reviewer 2, we encourage you to deem down your language throughout the manuscript on statements that are not supported by data.

Our general policy is that papers are considered through only one revision cycle; however, given that the suggested changes are relatively minor, we are open to one additional short round of revision. Please note that I will expect to make a final decision without additional reviewer input upon resubmission.

Please submit the final revision within one month, along with a letter that includes a point by point response to the remaining reviewer comments.

B. MANUSCRIPT ORGANIZATION AND FORMATTING:

Sincerely,

RE: Life Science Alliance Manuscript #LSA-2022-01439-TRR

Dr. Ogun Adebali
Sabanci University
Faculty of Engineering and Natural Sciences
Istanbul 34956
Turkey

Dear Dr. Adebali,

Thank you for submitting your revised manuscript entitled "Evolutionary association of receptor-wide amino acids with G protein coupling selectivity in GPCRs". We would be happy to publish your paper in Life Science Alliance pending final revisions necessary to meet our formatting guidelines.

- please consult our manuscript preparation guidelines <https://www.life-science-alliance.org/manuscript-prep> and make sure your manuscript sections are in the correct order;
- please make sure that you have a separate figure legend section
- please use the [10 author names, et al.] format in your references (i.e. limit the author names to the first 10)
- please add a callout for Figure 5 in the main manuscript text

To upload the final version of your manuscript, please log in to your account:
<https://lsa.msubmit.net/cgi-bin/main.plex>

A. FINAL FILES:

-- Summary blurb (enter in submission system): A short text summarizing in a single

sentence the study (max. 200 characters including spaces). This text is used in conjunction with the titles of papers, hence should be informative and complementary to the title. It should describe the context and significance of the findings for a general readership; it should be written in the present tense and refer to the work in the third person. Author names should not be mentioned.

B. MANUSCRIPT ORGANIZATION AND FORMATTING:

Sincerely,

4th Editorial Decision

13 May 2022

RE: Life Science Alliance Manuscript #LSA-2022-01439-TRRR

Dr. Ogun Adebali

Sabanci University
Faculty of Engineering and Natural Sciences
Istanbul 34956
Turkey

Dear Dr. Adebali,

Thank you for submitting your Research Article entitled "Evolutionary association of receptor-wide amino acids with G protein coupling selectivity in GPCRs". It is a pleasure to let you know that your manuscript is now accepted for publication in Life Science Alliance. Congratulations on this interesting work.

DISTRIBUTION OF MATERIALS:

Again, congratulations on a very nice paper. I hope you found the review process to be constructive and are pleased with how the manuscript was handled editorially. We look forward to future exciting submissions from your lab.

Sincerely,
